# An Objective Structured Clinical Examination (OSCE) for French Dental Students: Feedback after 2 Years

**DOI:** 10.3390/dj9110136

**Published:** 2021-11-19

**Authors:** Claire Egloff-Juras, Pierre Hirtz, Amandine Luc, Anne-Sophie Vaillant-Corroy

**Affiliations:** 1Faculté d’Odontologie, Université de Lorraine, F-54505 Vandoeuvre-lès-Nancy, France; pierre.hirtz@univ-lorraine.fr (P.H.); anne-sophie.vaillant@univ-lorraine.fr (A.-S.V.-C.); 2Service d’Odontologie, CHRU-Nancy, F-54000 Nancy, France; 3Université de Lorraine, CNRS UMR 7039, CRAN, F-54000 Nancy, France; 4Platform of Clinical Research Support PARC (MDS Unity), University Hospital of Nancy, F-54500 Vandoeuvre-lès-Nancy, France; a.luc@chru-nancy.fr

**Keywords:** Objective Structured Clinical Examination (OSCE), predoctoral dental education

## Abstract

The Objective Structured Clinical Examination (OSCE) is a practical examination that provides a standardized assessment of clinical competence. The aim of this study is to evaluate the objectivity and the reliability of an OSCE in dentistry. To this end, a retrospective monocentric observational study was conducted at the Faculty of Dentistry of Nancy by analyzing the exam results of 81 students. The study population consisted of the fifth-year students. The examination was broken down into six stations which were doubled, and different juries of examiners were constituted (installed in different rooms) according to the same composition. The *p*-value was set at 0.05. We found an equivalence of the results between the different rooms on the global mean score obtained at the six stations (*p* = 0.021). In terms of gender, women have statistically significantly higher overall scores than men (*p* = 0.001). The evaluation of a difference in the scores between full-time and part-time teachers does not find any statistically significant difference or equivalence in the station where it was possible to realize the comparison. However, the students’ waiting time before the exam seems to negatively influence the results. Compared with other international OSCE studies, the results presented seemed sufficiently objective and reliable, although some adjustments are still necessary.

## 1. Introduction

In 2014, the teaching team of the Faculty of Odontology of Nancy decided to reform the Certificate of Clinical and Therapeutics Synthesis, an examination which validated the fifth-year students and gave them the right to replace and to prescribe. Up to that date, it involved a written evaluation and a double oral evaluation (one jury for adult care and one for child care). It was then decided to replace the oral part with an Objective Structured Clinical Examination (OSCE) test. 

The OSCE is a practical examination that provides a standardized assessment of clinical competence [1]. Since the OSCE formula was first described by Harden in 1975 [2], it has been the subject of numerous studies and has been widely adopted by educational institutions, particularly in the health sciences (Medicine, Pharmacy, Odontology) [3,4,5,6]. The OSCE tests are today considered as the gold standard for the evaluation of clinical competence [7,8]. OSCEs are widely used in the assessment of dental students and have proven to be very effective [9]. These are multi-station tests, which use standardized, real, or simulated patients. The OSCE assesses the clinical skills, attitudes, or aptitudes of a candidate for a discipline. 

This examination should be clinical and should measure the skills and clinical performance of the candidates. The contexts of the evaluation are standardized and the candidate is only asked about the content of the station. However, it also has to be objective; therefore, all candidates are evaluated using the same stations with the same grid and the same rating scheme. Finally, it must be structured: each station of the examination has a very specific objective. When simulated patients are required, very detailed scripts are provided to ensure that the information given to the candidates is the same, including the emotions that the patients must manifest during the consultation.

The OSCE consists of a succession of examination stations; the duration of each station may vary but all the stations of a same examination must have the same duration. For each station, the candidate receives a brief written statement of a clinical problem and has a few min alone in a different room to prepare his response. The candidate is observed and evaluated by an examiner, who uses a standardized correction grid for this purpose. Each station has a different examiner, contrary to a traditional method of clinical examinations where a candidate could be assigned to the same jury for the full exam. All candidates realize the same stations and the stations are standardized, allowing comparison between the candidates [2].

The number of candidates evaluated in an OSCE varies from 5 to 1737; however, more frequently, there is an average of 50 candidates. The number of stations set up for this type of examination may vary from 2 to 52. However, the most common range is between 6 and 10 stations. The duration of each station oscillates between 3 and 40 min, with an average duration between 5 and 9 min [2,10,11].

One of the main qualities of the OSCE is objectivity: the stations are identical for all candidates, who have the same time to complete their station. The objectivity of the OSCE is similar to that of other examinations, such as multiple choice tests [8]. However, this objectivity is correlated with some variables related to the implementation of the OSCE. It increases with the number of stations, with the standardization of patients and with the number of evaluators. It decreases with the students’ fatigue and anxiety [12].

Nevertheless, the complexity of its implementation may be the limit. Compared to conventional oral exams or other examinations, the OSCE represents a much larger investment in terms of time and human means [8,13].

The introduction of OSCEs has completely reformed the end-of-cycle evaluation of our students. This is a new way of conducting oral examinations which requires learning but also adaptations according to the results obtained. We, therefore, felt it was necessary to verify that the OSCE modalities chosen for our exam were in accordance with the standards of this type of exam. The aim of this study is, thus, to evaluate the objectivity and the reliability of an OSCE in dentistry.

## 2. Materials and Methods

A retrospective monocentric observational study was conducted at the Faculty of Dentistry of Nancy by analyzing the exam results of 81 students. The study population consisted of the fifth-year students who must undergo an OSCE for their final exam. All students enrolled in the examination were recruited in this study. As this research project does not fall within the scope of the Committee for the Protection of Persons, it was submitted to the Ethics Committee of the University Hospital of Nancy, which gave a favorable opinion on the implementation of this study. The examination was broken down into 4 stations without the use of dental equipment and 2 stations in the care unit involving the manipulation of dental equipment. The 6 stations were doubled, meaning that 2 juries of examiners were constituted according to the same composition. All students were called at the same time in the morning and waited together in the same room. At their scheduled time, they went to either exam room 1 or exam room 2. In each exam room, different juries were present representing the different stations. The students were randomized between the 2 examination rooms.

For the script of our stations, we followed one important guideline: the reason for the consultation should be frequent without being banal (no complex consultation, only one clear and unambiguous motive). All scripts were tested twice by panels of teachers (both internal and external to the station discipline). The disciplines evaluated during the stations without equipment were pediatric dentistry, prosthodontics, conservative dentistry, and emergency care. In the one with equipment, it was prosthodontics and hygiene.

The data recorded were the scores obtained by the students sorted according to the examination room, according to the stations, according to the sex of the student, according to their waiting time between their arrival at 8 am and their running order, and according to whether the evaluator was a full-time or part-time teacher; and finally the students’ predictive scores after their examination.

The main objective of this study was to verify the equivalence of the results obtained by the students in the 2 rooms and, thus, to check the reproducibility of the examination according to the jury.

The secondary objectives were to compare the equivalence of the results obtained according to the different parameters (full-time or part-time teachers, students’ sex, waiting times) and to explore the self-assessment capacities of the students.

Characteristics of sample were described by percentage for categorical variables and means and standard deviation values for continued variables. The Student’s *t*-test was used to compare marks when they followed a normal distribution and the Wilcoxon test was used otherwise. If there was no difference in the previous tests, the equivalence of marks was evaluated by two one-sided tests (TOST) with an equivalence margin set at 1 point. The Wilcoxon signed-rank test was used to determine the evaluation of the self-assessment. The correlation between the marks and the waiting time was obtained with the Spearman coefficient. The *p*-value was set at 0.05.

These statistical analyses were performed using Statistical Analysis Software (SAS) version 9.4 (SAS Institute Inc., Cary, NC, USA).

## 3. Results

Eighty-one students (thirty-four men and forty-seven women) registered at the Dental Faculty of Nancy were included and divided into two groups corresponding to the two exam rooms (with 39 students in one room and 42 in the other).

The marks followed a normal distribution in these stations: prosthodontics with and without material and emergency station (which was not the case in the pediatric, conservative, and hygiene stations).

A significant difference was observed in the mean scores between the two rooms only for pediatric dentistry. For the other five stations, there was no significant difference between the two rooms (Table 1).

Tests of equivalence were carried out in these stations, and an equivalence was found in the results between the two rooms only on the overall mean obtained at the six stations (*p* = 0.021). In terms of gender, women had statistically significantly higher overall scores than men (*p* = 0.001). The search for a difference in the scores between full-time and part-time teachers did not find any statistically significant difference or equivalence. The students’ waiting time before the exam seemed to influence the results. Indeed, there was a weak but significant correlation (Spearman coefficient = −0.29, *p* = 0.010) between these two parameters. The Spearman coefficient being negative means that the longer the waiting time leads to a lower score. Finally, there was a statistically significant difference between the scores obtained and estimated for the prosthetic stations without equipment and the restorative dentistry with scores significantly higher than the estimated scores, and for the hygiene station with scores significantly lower than the estimated scores (Table 2).

## 4. Discussion

The implementation of OSCE stations for the evaluation of odontology students turned out to be interesting in many respects.

Our format of doubled stations on two rooms proved to be reproducible since we observed an equivalence of the mean marks obtained in the two rooms. So, we were able to confirm the objectivity of this type of examination. For these exams, we chose to make stations of 6 min with 1 min of change between the stations, which seemed perfectly satisfactory. In the 2011 Ebberhard study [14], the stations had a duration of 4 min with 1 min of change between each student, as in that of Schoonheim [15]. Another study published in 2009 reported a duration of 5 min [16]. In the Ebberhard study of 2011, 11 stations were set up [14]. In Schoonheim [15,16], between 16 and 18 stations are reported. The authors concluded that a minimum of 12 stations was required to validate the reliability of the examination. In our case, we had to reduce the number of stations to 6, due to the small number of teachers available. The validity of our examination can, therefore, be discussed.

In the Park study published in 2015, part-time teachers tended to give higher marks than full-time teachers [17]. Whereas, in our study, this does not seem to be the case, with an insignificant difference, but also an absence of equivalence. Nevertheless, after these 2 years, we validated and optimized our procedure of setting up OSCE stations which is today perfectly improved.

To ensure the secrecy of the subjects, the students were all summoned for the beginning of the first test and had to wait in a supervised room until it was their turn. Nevertheless, this seemed problematic since we found that the longer waiting times for students led to worse results. It would, therefore, be necessary to make adjustments, such as triple or even quadruple the stations instead of only doubling them in order to limit the students’ waiting time. Another possibility would be to call students at their exact exam time and have them wait in a monitored room with no means of communication after taking their exam. However, this last option may be difficult to get students to accept. This decrease in results when the waiting time increases can easily be explained by the increase in student stress related to this prolonged waiting time. Moreover, it is known that OSCE examinations are in any case a source of great stress for students [18]. Germa et al. [19] proposed a serious game to train for OSCE, achieving an interesting reduction in the stress level of the students through this.

Increasing the number of rooms raises another problem: the lack of supervision and the increase in the cost of the examinations. Due to the COVID-19 pandemic, different teams have proposed the realization of online OSCE with satisfactory results for both students and teachers. This modality could also be an answer to this logistic problem [20,21].

Indeed, the Eberhard study concluded that the total cost of their examination in the form of OSCE was 181 euros per student [14], whereas a German study found a cost per student of 161 euros [22,23]. In our case, as all the people who took part in the examination were employed by the faculty, there were no staff costs. On the other hand, the closure of the clinical service, as a result of the mobilization of all the teachers for the examinations, incurred a considerable cost. It should be noted that the studies cited also take into account the number of hours spent preparing for this examination.

Another difficulty encountered in the implementation of these stations was to obtain a consensus among the teachers as to the treatment proposed, the additional examinations to be carried out, and even the diagnosis to be made. The Whelan and Gerald study, published in 2005, presented the same problems [24]. This should lead us to rethink our lessons and to validate all the scoring grids within a teaching team.

## 5. Conclusions

After two years of implementation, the OSCE tested at the Faculty of Odontology of Nancy seemed sufficiently adapted to show objectivity, reproducibility, and efficiency for the evaluation of students in dentistry. The use of short stations (about 6 min) seemed plausible in these results. Even though a high number of stations is usually recommended in other studies (around 12 stations), our six-station format still allowed objectivity and easier implementation of the assessment.

The development of scripts and evaluation grids that are as precise as possible makes it possible to erase the differences in ratings that are sometimes observed, particularly between full-time and part-time teachers.

Compared with other international OSCE studies, the results presented seemed sufficiently objective and reliable, although some adjustments are still necessary. Our choice of calling all students at the same time and making them wait under supervision until their exam time does not seem satisfactory, as grades were significantly and negatively influenced by the increase in students’ waiting time.

## Figures and Tables

**Table 1 dentistry-09-00136-t001:** Comparison of scores between the two examination rooms.

	Room A			Room B			
	*N* = 42 (51.9%)			*N* = 39 (48.1%)			
	*N*	Mean	SD *	*N*	Mean	SD *	*p* **
**Without equipment station**							
**Pediatric Dentistry**	42	12.3	3.4	39	13.7	4.2	**0.014**
**Prosthodontics**	42	13.2	2.8	39	14.5	3.1	0.055
**Restorative Dentistry**	42	10.9	4.0	39	10.4	3.2	0.129
**Emergency**	42	9.4	3.6	39	8.7	2.6	0.346
**With equipment station**							
**Prosthodontics**	42	12.0	2.6	39	11.6	2.6	0.539
**Hygiene**	42	10.5	5.5	39	8.9	5.2	0.271
**Overall mean**	42	11.4	2.2	39	11.3	1.9	0.881

* Standard deviation ** Student’s *t*-test or Wilcoxon test.

**Table 2 dentistry-09-00136-t002:** Comparison of obtained and self-evaluated scores.

	Obtained			Self-Evaluated			
	*N*	Mean	SD *	*N*	Mean	SD *	*p* **
**Without equipment station**							
**Pediatric dentistry**	81	13.0	3.8	81	12.6	2.6	**0.080**
**Prosthodontics**	81	13.8	3.0	81	12.0	2.3	**<0.0001**
**Restorative dentistry**	81	10.6	3.6	81	8.7	2.8	**<0.0001**
**Emergency**	81	9.0	3.2	81	9.5	2.5	0.363
**With equipment station**							
**Prosthodontics**	81	11.8	2.6	81	11.4	2.5	0.114
**Hygiene**	81	9.7	5.4	81	11.7	2.2	**0.002**
**Overall mean**	81	11.3	2.0	81	11.0	1.3	0.098

* Standard deviation ** Wilcoxon signed-rank test.

## Data Availability

The data presented in this study are available on request from the corresponding author.

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
