# Peer review of "An Objective Structured Clinical Examination (OSCE) for French Dental Students: Feedback after 2 Years"

_dentistry, 2021, doi:10.3390/dj9110136_

Round 1

Reviewer 1 Report

Dear authors,

The article "Objective Structured Clinical Examination (OSCE) in dentistry: feedback after 2 years of setting up", may be of interest to certain categories of scholars. 

The article is clear and well written, the information is relevant for the subject.

I have no comments or suggestions, in my opinion, the article may be published in its current form. 

Author Response

Thank you very much for the positive feedback and for taking the time to review our work.

Reviewer 2 Report

The authors have attempted a very good topic. The authors are suggested to do the following minor revisions.

1) Introduction can be strengthened by incorporating the justification of doing the study with more details.

2) The figures are not representing any data and therefor recommended to add at least one figure or graphical presentation.

3) Do the OSCE scripts vetted? If yes who were the panel members and what criteria was set to ascertain their validity and reliability. 

4) Please search the literature vigorously. You can find relevant references to add to your manuscript.

Author Response

The authors have attempted a very good topic. The authors are suggested to do the following minor revisions.

First, thank you very much for the positive feedback and for taking the time to read our work.

1) Introduction can be strengthened by incorporating the justification of doing the study with more details.

The following sentence has been added at the end of the introduction: “The introduction of OSCEs has completely reformed the end-of-cycle evaluation of our students. This is a new way of conducting oral examinations which requires learning but also adaptations according to the results obtained. We therefore felt it was necessary to verify that the OSCE modalities chosen for our exam were in accordance with the standards of this type of exam.”

2) The figures are not representing any data and therefor recommended to add at least one figure or graphical presentation.

We have chosen to present all the results in the form of tables because it seemed to us more readable and clearer. Unfortunately, we are not sure that we understand the type of figure requested here. If you could explain to us what is expected, we could then add the requested figure.

3) Do the OSCE scripts vetted? If yes who were the panel members and what criteria was set to ascertain their validity and reliability.

All scripts were tested twice by a panel of faculty members. The scripts were analyzed by both discipline teachers and non-discipline teachers. For this, the scripts were tested with a non-discipline teacher in the role of the student and a panel of teachers observing the station in order to analyze its content. After the stations were corrected, a new simulation was carried out.

The following sentence has been added to the manuscript in the material and method section: “All scripts were tested twice by panels of teachers (both internal and external to the station discipline).”

4) Please search the literature vigorously. You can find relevant references to add to your manuscript.

5 recent bibliographic references have been added.

One in the introduction:

- “OSCEs are widely used in the assessment of dental students and have proven their ef-fectiveness” (9).

4 in the discussion:

- This decrease in results when the waiting time increases can easily be explained by the increase in student stress related to this prolonged waiting time. Moreover, it is known that OSCE examinations are in any case a source of great stress for students (18). Germa et al. (19) proposed a serious game to train for OSCE, achieving an interesting reduction in the stress level of the students through this.

- Due to the Covid 19 pandemic, different teams have proposed the realization of online OSCE with satisfactory results for both students and teachers. This modality could also be an answer to this logistic problem (20,21).

Reviewer 3 Report

The manuscript is interesting and it could be accepted 

Author Response

(The authors gave the same response as above.)

Reviewer 4 Report

It is an interesting assessment of OSCE exam, but some issues are unclear. 

Authors claim that "the examination was broken down into 6 stations which were doubled or tripled" (abstract), but they do not mention in material and methods which stations were tripled.

Wilcoxon test was used to compare results from two rooms, but different subgroups of students took exam in these rooms, so I am not sure if we can treat them as matched samples. Shouldn't we use Mann-Whitney test instead?

I do not understand how the students rotated between stations. And when the had to wait: before? in the meantime? during rest stations?

Did they take the stations in the same order or everone started from a different station? It would be interesting to check if running order had impact on the results.

Authors propose "Another possibility would be to call students at their 
exact exam time and have them wait in a monitored room with no means of communication after taking their exam. However, this last option may be difficult to get students to accept". How was it done in fact? Weren't they call at the exact time? Didn't they wait in a monitored room?

Author Response

It is an interesting assessment of OSCE exam, but some issues are unclear.

Thank you very much for the positive feedback and for taking the time to review our work.

  1. Authors claim that "the examination was broken down into 6 stations which were doubled or tripled" (abstract), but they do not mention in material and methods which stations were tripled.

This is an error in the abstract. All stations have been doubled. The correction has been made in the abstract.

  1. Wilcoxon test was used to compare results from two rooms, but different subgroups of students took exam in these rooms, so I am not sure if we can treat them as matched samples. Shouldn't we use Mann-Whitney test instead?

According to our bio statistician, the Wilcoxon test is the most suitable in this case.

  1. I do not understand how the students rotated between stations. And when they had to wait: before? in the meantime? during rest stations?

All students are called at the same time in the morning and wait together in the same room. At their scheduled time, they go to either exam room 1 or exam room 2. In each exam room, different juries are present representing the different stations. They pass on all 4 stations without a break between the 4 stations. Details have been provided in the manuscript.

“All students are called at the same time in the morning and wait together in the same room. At their scheduled time, they go to either exam room 1 or exam room 2. In each exam room, different juries are present representing the different stations” added in material &methods.

  1. Did they take the stations in the same order or everone started from a different station? It would be interesting to check if running order had impact on the results.

Indeed, they pass all the stations in a different order. It would therefore seem very interesting to look for an influence of the order in which the stations pass. We will include this parameter in our next analysis of the results.

  1. Authors propose "Another possibility would be to call students at their

exact exam time and have them wait in a monitored room with no means of communication after taking their exam. However, this last option may be difficult to get students to accept". How was it done in fact? Weren't they call at the exact time? Didn't they wait in a monitored room?

Currently they are all called at the same time in the morning and they wait BEFORE taking their exam. The idea would be to make them wait AFTER their exam to limit the impact of stress that increases as they wait.
